# Backprop with Approximate Activations for Memory-efficient Network Training

**Ayan Chakrabarti**
Washington University in St. Louis
1 Brookings Dr., St. Louis, MO 63130
ayan@wustl.edu

**Benjamin Moseley**
Carnegie Mellon University
5000 Forbes Ave., Pittsburgh, PA 15213
moseleyb@andrew.cmu.edu

## Abstract

Training convolutional neural network models is memory intensive since back-propagation requires storing activations of all intermediate layers. This presents a practical concern when seeking to deploy very deep architectures in production, especially when models need to be frequently re-trained on updated datasets. In this paper, we propose a new implementation for back-propagation that significantly reduces memory usage, by enabling the use of approximations with negligible computational cost and minimal effect on training performance. The algorithm reuses common buffers to temporarily store full activations and compute the forward pass exactly. It also stores approximate per-layer copies of activations, at significant memory savings, that are used in the backward pass. Compared to simply approximating activations within standard back-propagation, our method limits accumulation of errors across layers. This allows the use of much lower-precision approximations without affecting training accuracy. Experiments on CIFAR-10, CIFAR-100, and ImageNet show that our method yields performance close to exact training, while storing activations compactly with as low as 4-bit precision.

## 1 Introduction

The use of deep convolutional neural networks has become prevalent for a variety of visual and other inference tasks [9], with a trend to employ increasingly larger and deeper network architectures [8, 10] that are able to express more complex functions. While deeper architectures have delivered significant improvements in performance, they have also increased demand on computational resources. In particular, training such networks requires a significant amount of on-device GPU memory—much more so than during inference—in order to store activations of all intermediate layers of the network that are needed for gradient computation during back-propagation.

This leads to large memory footprints during training for state-of-the-art deep architectures, especially when training on high-resolution images with a large number of activations per layer. This in-turn can lead to the computation being inefficient and "memory-bound": it forces the use of smaller training batches for each GPU leading to under-utilization of available GPU cores (smaller batches also complicate the use of batch-normalization [12] with batch statistics computed over fewer samples). Consequently, practitioners are forced to either use a larger number of GPUs for parallelism, or contend with slower training. This makes it expensive to deploy deep architectures for many applications, especially when networks need to be continually trained as more data becomes available.

Prior work to address this has traded-off memory for computation [2, 4, 5, 14], but with a focus on enabling exact gradient computation. However, stochastic gradient descent (SGD) inherently works with noisy gradients at each iteration and, in the context of distributed training, has been shown to succeed when using approximate and noisy parameter gradients when aggregating across multiple devices [3, 16, 19, 20]. Motivated by this, we propose a method that uses approximate activations

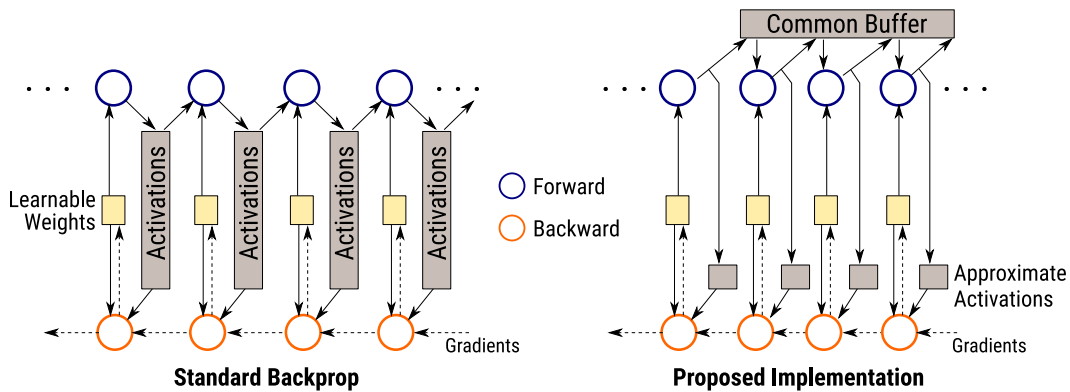

Figure 1: Proposed Approach. Standard backprop requires storing activations of each layer for gradient computation during the backward pass, leading to a large memory footprint. Our method permits these activations to be approximated to lower memory usage, while preventing errors from building up across layers. A common reusable buffer temporarily stores the exact activations for each layer during the forward pass, while retaining an approximate copy for the backward pass. Our method ensures the forward pass is exact and limits errors in gradients flowing back to earlier layers.

to significantly reduce memory usage on each GPU during training. Our method has virtually no additional computational cost and, since it introduces only a modest approximation error in computed parameter gradients, has minimal effect on training performance.

A major obstacle to using approximate activations for training is that, with the standard implementation of back-propagation (backprop), approximation errors *compound* across the forward and then backward pass through all layers. This limits the degree of approximation at each layer (e.g., some libraries support 16- instead of 32-bit floats for training). In this work, we propose a new backprop algorithm based on the following key observations: the forward pass through a network can be computed exactly without storing intermediate activations, and approximating activations has minimal effect on gradients "flowing back" to the input of a layer in the backward pass.

For the forward pass during training, our method uses a layer's exact activations to compute those for a subsequent layer. However, once they have been used, it overwrites the exact activations to reuse memory, and retains only an approximate copy with a lower memory footprint. During the backward pass, gradients are computed based on these stored approximate copies. This incurs only a modest error at each layer when computing parameter gradients due to multiplying the incoming gradient with the approximate activations, but ensures the error in gradients propagating back to the previous layer is minimal. Experiments show that our method can permit the use of a large degree of approximation without any significant degradation in training quality. This substantially lowers memory usage (by up to a factor of 8) during training *without requiring additional expensive computation*, thus making network training efficient and practical for larger and deeper architectures.

## 2 Related Work

A number of works focus on reducing the memory footprint of a model during inference, e.g., by compression [7] and quantization [11], to ensure that it can be deployed on resource-limited mobile devices, while still delivering reasonable accuracy. However, these methods do not address the significant amount of memory required to train the networks themselves (note [7, 11] require storing full-precision activations during training), which is significantly larger than what is needed for inference. The most common recourse to alleviate memory bottlenecks during training is to simply use multiple GPUs. But this often under-utilizes the available parallelism on each device—computation in deeper architectures is distributed more sequentially and, without sufficient data parallelism, often does not saturate all GPU cores—and also adds the overhead of intra-device communication.

A popular strategy to reduce training memory requirements is "checkpointing". Activations for only a subset of layers are stored at a time, and the rest recovered by repeating forward computations [2, 5, 14]. This affords memory savings with the trade-off of additional computational cost—e.g., [2]

propose a strategy that requires memory proportional to the square-root of the number of layers. However, it requires additional computation, with cost proportional to that of an additional forward pass. In a similar vein, [4] considered network architectures with "reversible" or invertible layers to allow re-computing input activations of such layers from their outputs during the backward pass.

These methods likely represent the best possible solutions if the goal is restricted to computing exact gradients. But SGD is fundamentally a noisy process, and the exact gradients computed over a batch at each iteration are already an approximation—of gradients of the model over the entire training set [17]. Researchers have posited that further approximations are possible without degrading training ability. For distributed training, asynchronous methods [3, 16] delay synchronizing models across devices to mitigate communication latency. Despite each device now working with stale models, there is no major degradation in training performance. Other methods quantize gradients to two [19] or three levels [20] so as to reduce communication overhead, and again find that training remains robust to such approximation. Our work also adopts an approximation strategy to gradient computation, but targets the problem of memory usage on a each device. We approximate activations, rather than gradients, in order to achieve significant reductions in a model's memory footprint during training. Moreover, since our method makes the underlying backprop engine more efficient, for any group of layers, it can also be used within checkpointing to further improve memory cost.

It is worth differentiating our work from those that carry out all training computations at lower-precision [1, 6, 15]. This strategy allows for a modest lowering of precision—from 32- to 16-bit representations, and to 8-bit with some loss in training quality for [1]—beyond which training error increases significantly. In contrast, our approach allows for much greater approximation by limiting accumulation of errors across layers, and we are able to replace 32-bit floats with 8- and even 4-bit fixed-point approximations, with little effect on training performance. Of course, performing all computation at lower-precision also has a computational advantage: due to reduction in-device memory bandwidth usage (transferring data from global device memory to registers) in [15], and due to the use of specialized hardware in [6]. While the goal of our method is different, it can also be combined with these ideas: compressing intermediate activations to a greater degree, while using 16-bit precision for computational efficiency.

## 3 Proposed Method

We now describe our approach to memory-efficient training. We begin by reviewing the computational steps in the forward and backward pass for a typical network layer, and then describe our approximation strategy to reducing the memory requirements for storing intermediate activations.

### 3.1 Background

A neural network is composition of linear and non-linear functions that map the input to the final desired output. These functions are often organized into "layers", where each layer consists of a single linear transformation—typically a convolution or a matrix multiply—and a sequence of non-linearities. We use the "pre-activation" definition of a layer, where we group the linear operation with the non-linearities that immediately *preceed* it. Consider a typical network whose $l^{th}$ layer applies batch-normalization and ReLU to its input $A_{l:i}$ followed by a linear transform:

$$\textbf{[B.Norm.]} \quad A_{l:1} = (\sigma_l^2 + \epsilon)^{-1/2} \circ (A_{l:i} - \mu_l), \quad \mu_l = \text{Mean}(A_{l:i}), \sigma^2 = \text{Var}(A_{l:i}); \quad (1)$$

$$\textbf{[Sc.\&B.]} \quad A_{l:2} = \gamma_l \circ A_{l:1} + \beta_l; \quad (2)$$

$$\textbf{[ReLU]} \quad A_{l:3} = \max(0, A_{l:2}); \quad (3)$$

$$\textbf{[Linear]} \quad A_{l:o} = A_{l:3} \times W_l; \quad (4)$$

to yield the output activations $A_{l:o}$ that are fed into subsequent layers. Here, each activation is a tensor with two or four dimensions: the first indexing different training examples, the last corresponding to "channels", and others to spatial location. $\text{Mean}(\cdot)$ and $\text{Var}(\cdot)$ aggregate statistics over batch and spatial dimensions, to yield vectors $\mu_l$ and $\sigma_l^2$ with *per-channel* means and variances. Element-wise addition and multiplication (denoted by $\circ$) are carried out by "broadcasting" when the tensors are not of the same size. The final operation represents the linear transformation, with $\times$ denoting matrix multiplication. This linear transform can also correspond to a convolution.

Note that (1)-(4) are defined with respect to learnable parameters $\gamma_l, \beta_l$, and $W_l$, where $\gamma_l, \beta_l$ are both vectors of the same length as the number of channels in $A_l$, and $W_l$ denotes a matrix (for

fully-connected layers) or elements of a convolution kernel. These parameters are learned iteratively using SGD, where at each iteration, they are updated based on gradients—$\nabla \gamma_l, \nabla \beta_l$, and $\nabla W_l$—of some loss function computed on a batch of training samples.

To compute gradients with respect to all parameters for all layers in the network, the training algorithm first computes activations for all layers in sequence, ordered such that each layer in the sequence takes as input the output from a previous layer. The loss is computed with respect to activations of the final layer, and then the training algorithm goes through all layers again in reverse sequence, using the chain rule to back-propagate gradients of this loss. For the $l^{th}$ layer, given the gradients $\nabla A_{l:o}$ of the loss with respect to the output, this involves computing gradients $\nabla \gamma_l, \nabla \beta_l$, and $\nabla W_l$ with respect to the layer's learnable parameters, as well as gradients $\nabla A_{l:i}$ with respect to its input for further propagation. These gradients are given by:

$$\textbf{[Linear]} \ \ \nabla W = A_{l:3}^T \times (\nabla A_{l:o}), \ \ \nabla A_{l:3} = (\nabla A_{l:o}) \times W_l^T; \tag{5}$$

$$\textbf{[ReLU]} \ \ \nabla A_{l:2} = \delta(A_{l:2} > 0) \circ (\nabla A_{l:3}); \tag{6}$$

$$\textbf{[Sc.\&B.]} \ \ \nabla \beta_l = \text{Sum}(\nabla A_{l:2}), \ \ \nabla \gamma_l = \text{Sum}\left(A_{l:1} \circ (\nabla A_{l:2})\right), \ \ \nabla A_{l:1} = \gamma_l \circ \nabla A_{l:2}; \tag{7}$$

$$\textbf{[B.Norm.]} \ \ \nabla A_{l:i} = (\sigma_l^2 + \epsilon)^{-1/2} \circ \left[\nabla A_{l:1} - \text{Mean}(\nabla A_{l:1}) - A_{l:1} \circ \text{Mean}(A_{l:1} \circ \nabla A_{l:1})\right]; \tag{8}$$

where $\text{Sum}(\cdot)$ and $\text{Mean}(\cdot)$ again aggregate over all but the last dimension, and $\delta(A > 0)$ is a tensor the same size as $A$ that is 1 where the values in $A$ are positive, and 0 otherwise.

When the goal is to just compute the final output of the network, the activations of an intermediate layer can be discarded during the forward pass as soon as we finish processing the subsequent layer or layers that use it as input. However, we need to store all intermediate activations during training because they are needed to compute gradients during back-propagation: (5)-(8) involve not just the values of the incoming gradient, but also the values of the activations themselves. Thus, training requires enough available memory to hold the activations of all layers in the network.

## 3.2 Back-propagation with Approximate Activations

We begin by observing we do not necessarily need to store all intermediate activations $A_{l:1}, A_{l:2}$, and $A_{l:3}$ *within* a layer. For example, it is sufficient to store the activation values $A_{l:2}$ right before the ReLU, along with the variance vector $\sigma_l^2$ (which is typically much smaller than the activations themselves). Given $A_{l:2}$, we can reconstruct the other activations $A_{l:3}$ and $A_{l:3}$ needed in (5)-(8) using element-wise operations at negligible cost. Some libraries already use such "fused" layers to conserve memory, and we use this to measure memory usage for exact training.

Storing one activation tensor at full-precision for every layer still requires a considerable amount of memory. We therefore propose retaining an approximate low-precision version $\tilde{A}_{l:2}$ of $A_{l:2}$, that requires much less memory for storage, for use in (5)-(8) during back-propagation. As shown in Fig. 2, we use full-precision versions of all activations during the forward pass to compute $A_{l:o}$ from $A_{l:i}$ as per (1)-(4), and use $A_{l:2}$ to compute its approximation $\tilde{A}_{l:2}$. The full precision approximations are discarded (i.e., over-written) as soon they have been used—the intermediate activations $A_{l:1}, A_{l:2}, A_{l:3}$ are discarded as soon as the approximation $\tilde{A}_{l:2}$ and output $A_{l:o}$ have been computed ($A_{l:o}$ is itself discarded after it has been used by a subsequent layer). Thus, only the approximate activations $\tilde{A}_{l:2}$ and variance vector $\sigma_l^2$ for each layer are retained for back-propagation, where it is also used to compute corresponding approximate versions $\tilde{A}_{l:1}$ and $\tilde{A}_{l:3}$, in (5)-(8).

Our method allows for the use of any generic approximation strategy to derive $\tilde{A}_{l:2}$ from $A_{l:2}$ that leads to memory-savings, with the only requirement being that the approximation preserve the sign of these activations (as will be discussed below). In our experiments, we use quantization to $K$-bit fixed point representations as a simple and computationally-inexpensive approximation strategy to validate our method. However, we believe that future work on more sophisticated and data-driven approaches to per-layer approximation can yield even more favorable memory-performance trade-offs.

Specifically, given desired bit-size $K$ and using the fact that $A_{l:1}$ is batch-normalized and thus $A_{l:2}$ has mean $\beta_l$ and variance $\gamma_l^2$, we compute an integer tensor $\tilde{A}_{l:2}^*$ from $A_{l:2}$ as:

$$\tilde{A}_{l:2}^* = \text{Clip}_K\left(\lfloor A_{l:2} \circ 2^K (6 * \gamma_l)^{-1} \rfloor + 2^{K-1} - \lfloor \beta_l \circ 2^K (6 * \gamma_l)^{-1} \rfloor\right), \tag{9}$$

where $\lfloor \cdot \rfloor$ indicates the "floor" operator, and $\text{Clip}_K(x) = \max(0, \min(2^K - 1, x))$. The resulting integers (between 0 and $2^K - 1$) can be directly stored with $K$-bits. When needed during back-

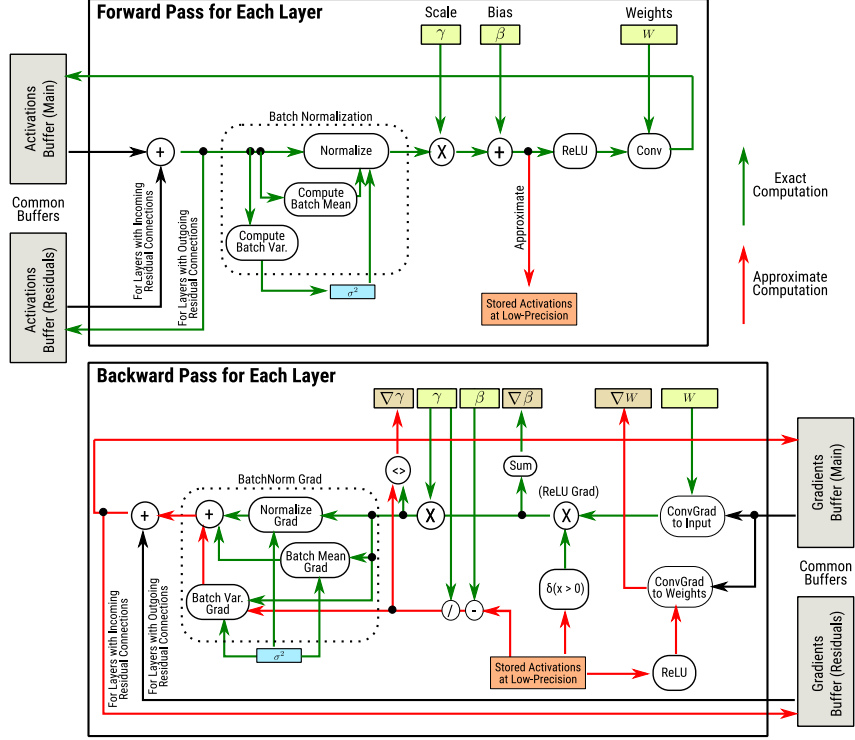

Figure 2: Details of computations involved in the forward and backward pass during network training with our method, for a single "pre-activation" layer with residual connections. We use two shared global buffers (for the straight and residual outputs) to store full-precision activations, ensuring the forward pass is exact. For each layer, we store approximate copies of the activations to save memory, and use these during back-propagation. Since our approximation preserves signs (needed to compute the ReLU gradient), most computation for the gradient back to a layer's input are exact—with only the backprop through the variance-computation in batch-normalization being approximate.

propagation, we recover a floating-point tensor holding the approximate activations $\tilde{A}_{l:2}$ as:

$$\tilde{A}_{l:2} = 2^{-K}(6 * \gamma_l) \circ \left(\tilde{A}_{l:2}^* + 0.5 - 2^{K-1} + \lfloor \beta_l \circ 2^K (6 * \gamma_l)^{-1} \rfloor\right) \qquad (10)$$

This simply has the effect of clipping $A_{l:2}$ to the range $\beta_l \pm 3\gamma_l$ (the range may be slightly asymmetric around $\beta_l$ because of rounding), and quantizing values in $2^K$ fixed-size intervals (to the median of each interval) within that range. It is easy to see that for values that are not clipped, the upper-bound on the absolute error between the true and approximate activations is $3/2^K \gamma_l$ for $A_{l:2}$ and $A_{l:3}$, and $3/2^K$ for $A_{l:1}$. Moreover, this approximation ensures that the sign of $A_{l:2}$ is preserved, i.e., $\delta(A_{l:2} > 0) = \delta(\tilde{A}_{l:2} > 0)$.

## 3.3 Approximation Error in Training

Since the forward computations happen in full-precision, there is no error introduced in any of the activations $A_l$ prior to approximation. To analyze errors in the backward pass, we examine how approximation errors in the activations affect the accuracy of gradient computations in (5)-(8). During the first back-propagation step in (5) through the linear transform, the gradient $\nabla W$ to the learnable transform weights will be affected by the approximation error in $\tilde{A}_{l:3}$. However, the gradient $\nabla A_{l:2}$ can be computed exactly (as a function of the incoming gradient to the layer $\nabla A_{l:o}$), because it does not depend on the activations. Back-propagation through the ReLU in (7) is also not affected, because it depends only on the sign of the activations, which is preserved by our approximation. When back-propagating through the scale and bias in (6), only the gradient $\nabla \gamma$ to the scale depends on the activations, but gradients to the bias $\beta_l$ and to $A_{l:1}$ can be computed exactly.

And so, while our approximation introduces some error in the computations of $\nabla W$ and $\nabla \gamma$, the gradient flowing towards the input of the layer is exact, until it reaches the batch-normalization

operation in (8). Here, we do incur an error, but note that this is only in one of the three terms of the expression for $\nabla A_{l:i}$—which accounts for back-propagating through the variance computation, and is the only term that depends on the activations. Hence, while our activation approximation does introduce some errors in the gradients for the learnable weights, we limit the accumulation of these errors across layers because a majority of the computations for back-propagation to the input of each layer are exact. This is illustrated in Fig. 2, with the use of green arrows to show computations that are exact, and red arrows for those affected by the approximation.

### 3.4 Network Architectures and Memory Usage

Our full training algorithm applies our approximation strategy to every layer (defined by grouping linear transforms with preceding non-linear activations) during the forward and backward pass. Skip and residual connections are handled easily, since back-propagation through these connections involves simply copying to and adding gradients from both paths, and doesn't involve the activations themselves. We assume the use of ReLU activations, or other such non-linearities such as "leaky"-ReLUs whose gradient depends only on the sign of the activations. Other activations (like sigmoid) may incur additional errors—in particular, we do not approximate the activations of the final output layer in classifier networks that go through a Soft-Max. However, since this is typically at the final layer for most convolutional networks, and computing these activations is immediately followed by back-propagating through that layer, approximating these activations offers no savings in memory. Note that our method has limited utility for architectures where a majority of layers have saturating non-linearities (as is the case for most recurrent networks).

Our approach also handles average pooling by simply folding it in with the linear transform. For max-pooling, exact back-propagation through the pooling operation would require storing the arg-max indices (the number of bits required to store these would depend on the max-pool receptive field size). However, since max-pool layers are used less often in recent architectures in favor of learned downsampling (ResNet architectures for image classification use max-pooling only in one layer), we instead choose not to approximate layers with max-pooling for simplicity.

Given a network with $L$ layers, our memory usage depends on connectivity for these layers. Our approach requires storing the approximate activations for each layer, each occupying reduced memory rate at a fractional rate of $\alpha < 1$. During the forward pass, we also need to store, at full-precision, those activations that are yet to be used by subsequent layers. This is one layer's activations for feed-forward networks, and two layers' for standard residual architectures. More generally, we will need to store activations for upto $W$ layers, where $W$ is the "width" of the architecture—which we define as the maximum number of outstanding layer activations that remain to be used as process layers in sequence—this width is one for simple feed-forward architectures and two for standard residual networks, but may be higher for DenseNet architectures [10]. During back-propagation, the same amount of space is required for storing gradients till they are used by previous layers. We also need space to re-create a layer's approximate activations as full-precision tensors from the low-bit stored representation, for use in computation.

Thus, assuming that all activations of layers are the same size, our algorithm requires $\mathcal{O}(W + 1 + \alpha L)$ memory, compared to the standard requirement of $\mathcal{O}(L)$. For our simple quantized fixed-point approximation strategy, this leads to substantial savings for deep networks with large $L$ since $\alpha = \frac{1}{4}, \frac{1}{8}$ when approximating 32-bit floating point activations with $K = 8, 4$ bits.

## 4 Experiments

In this section, we present experimental results which demonstrate that:

- Our algorithm enables up to 8x memory savings, with negligible drop in training accuracy compared to exact training, and significantly superior performance over other baselines.
- The lower memory footprint allows training to fully exploit available parallelism on each GPU, leading to faster training for deeper architectures.

We implement the proposed approximate memory-efficient training method as a general library that accepts specifications for feed-forward architectures with possible residual connections (i.e., $W = 2$). As illustrated in Fig. 2, it allocates a pair of common global buffers for the direct and residual paths. At any point during the forward pass, these buffers hold the full-precision activations

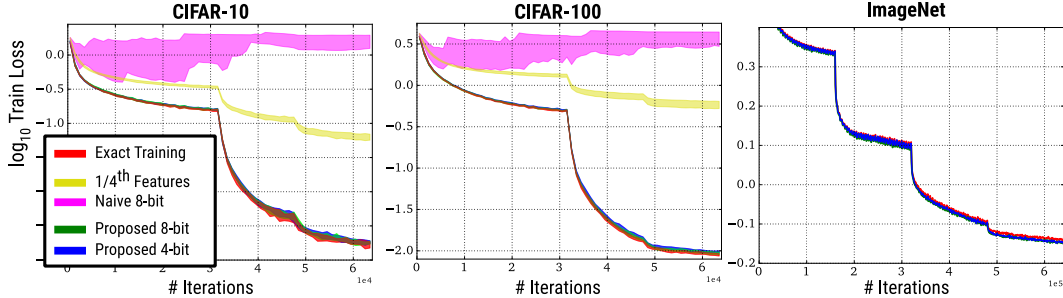

Figure 3: Approximate Training on CIFAR and ImageNet. We show the evolution of training losses for ResNet-164 models trained on CIFAR-10 and CIFAR-100, and ResNet-152 models trained on ImageNet with exact training and using our approximation strategy. CIFAR results are summarized across ten random initializations with bands depicting minimum and maximum loss values. We find that the loss using our method closely follow that of exact training through all iterations. For CIFAR, we also include results for training when using a "naive" 8-bit approximation baseline—where the approximate activations are also used in the forward pass. In this case, errors accumulate across layers and we find that training fails.

that are needed for computation of subsequent layers. The same buffers are used to store gradients during the back-ward pass. Beyond these common buffers, the library only stores the low-precision approximate activations for each layer for use during the backward-pass.

We compare our approximate training approach, with 8- and 4-bit activations, to exact training with full-precision activations. For fair comparison, we only store one set of activations (like our method, but with full precision) for a group of batch-normalization, ReLU, and linear (convolution) operations. As baselines, we consider exact training with fewer activations per-layer, as well as a naive approximation method with standard backprop. This second baseline replaces activations with low-precision versions directly during the forward pass. We do this conservatively and all computations are carried out in full precision. For each layer, the activations are only approximated right before the ReLU like in our method, but use this approximated-version as input to the subsequent convolution operation. Note that all methods use exact computation at test time.

**CIFAR-10 and CIFAR-100.** We begin with comparisons on 164-layer pre-activation residual networks [9] on CIFAR-10 and CIFAR-100 [13], using three-layer "bottlneck" residual units and parameter-free shortcuts for all residual connections. We train the network for 64k iterations with a batch size of 128, momentum of 0.9, and weight decay of $2 \times 10^{-4}$. Following [9], the learning rate is set to $10^{-2}$ for the first 400 iterations, then increased to $10^{-1}$, and dropped by a factor of 10 at 32k and 48k iterations. We use standard data-augmentation with random translation and horizontal flips. We train these networks with our approach using $K = 8$ and $K = 4$ bit approximations, and measure degradation in accuracy with respect to exact training—repeating training for all cases with ten random seeds. We also include comparisons to exact training with $1/4^{th}$ the number of feature channels in each layer, and to the naive approximation baseline (with $K = 8$ bits, i.e., $\alpha = 1/4$).

We visualize the evolution of training losses in Fig. 3, and report test set errors of the final model in Table 1. We find that the training loss when using our low-memory approximation strategy closely follow those of exact back-propagation, throughout the training process. Moreover, the final mean test errors of models trained with even 4-bit approximations (i.e., corresponding to $\alpha = 1/8$) are only slightly higher than those trained with exact computations, with the difference being lower than the standard deviation across different initializations. In contrast, training with fewer features per layer leads to much lower accuracy, while the naive approximation baseline simply fails, highlighting the importance of preventing accumulation of errors across layers using our approach.

**ImageNet.** For ImageNet [18], we train models with a 152-layer residual architecture, again using three-layer bottleneck units and pre-activation parameter-free shortcuts. We use a batch size of 256 for a total of 640k iterations with a momentum of 0.9, weight decay of $10^{-4}$, and standard scale, color, flip, and translation augmentation. The initial learning rate is set to $10^{-1}$ with drops by factor of 10 every 160k iterations. Figure 3 shows the evolution of training loss in this case as well, and Table 1 reports top-5 validation accuracy (using 10 crops at a scale of 256) for models trained using

Table 1: Accuracy Comparisons on CIFAR-10, CIFAR-100, and ImageNet with 164- (for CIFAR-10 and CIFAR-100) and 152- (for ImageNet) layer ResNet architectures. CIFAR results show mean ± std over training with ten random initializations for each case. ImageNet results use 10-crop testing.

| | | CIFAR-10<br>Test Set Error | CIFAR-100<br>Test Set Error | ImageNet<br>Val Set Top-5 Error |
|---|---|---|---|---|
| Exact | $(\alpha = 1)$ | 5.36%±0.15 | 23.44%±0.26 | 7.20% |
| Exact w/ fewer features | $(\alpha = 1/4)$ | 9.49%±0.12 | 33.47%±0.50 | - |
| Naive 8-bit Approx. | $(\alpha = 1/4)$ | 75.49%±9.09 | 95.41%±2.16 | - |
| *Proposed Method* | | | | |
| 8-bit | $(\alpha = 1/4)$ | 5.48%±0.13 | 23.63%±0.32 | 7.70% |
| 4-bit | $(\alpha = 1/8)$ | 5.49%±0.16 | 23.58%±0.30 | 7.72% |

Table 2: Comparison of maximum batch-size and wall-clock time per training example (i.e., training time per-iteration divided by batch size) for different ResNet architectures on CIFAR-10.

| # Layers | | 1001 (4x) | 1001 | 488 | 254 | 164 |
|---|---|---|---|---|---|---|
| Maximum | Exact | 26 | 134 | 264 | 474 | 688 |
| Batch-size | 4-bit | **182** | **876** | **1468** | **2154** | **2522** |
| Run-time | Exact | 130.8 ms | 31.3 ms | 13.3 ms | **6.5 ms** | **4.1 ms** |
| per Sample | 4-bit | **101.6 ms** | **26.0 ms** | **12.7 ms** | 6.7 ms | 4.3 ms |

exact computation, and our approach with $K = 8$ and $K = 4$ bit approximations. As with the CIFAR experiments, training losses using our strategy closely follow that of exact training (interestingly, the loss using our method is slightly lower than that of exact training during the final iterations, although this is likely due to random initialization), and the drop in validation set accuracy is again relatively small: at $0.5\%$ for a memory savings factor of $\alpha = 1/8$ with $K = 4$ bit approximations.

**Memory and Computational Efficiency.** For the CIFAR experiments, we were able to fit the full 128-size batch on a single 1080Ti GPU for both exact training and our method. For ImageNet training, we parallelized computation across two GPUs, and while our method was able to fit half a batch (size 128) on each GPU, exact training required two forward-backward passes (followed by averaging gradients) with $64-$sized batches per-GPU per-pass. In both cases, the per-iteration run-times were nearly identical. However, these represent comparisons restricted to having the same total batch size (needed to evaluate relative accuracy). For a more precise evaluation of memory usage, and the resulting computational efficiency from parallelism, we considered residual networks for CIFAR-10 of various depths up to 1001 layers—and additionally for the deepest network, a version with four times as many feature channels in each layer. For each network, we measured the largest batch size that could be fit in memory with our method (with $K = 4$) vs exact training, i.e., $b$ such that a batch of $b + 1$ caused an out-of-memory error on a 1080Ti GPU. We also measured the corresponding wall-clock training time per sample, computed as the training time per-iteration divided by this batch size. These results are summarized in Table 2. We find that in all cases, our method allows significantly larger batches to be fit in memory. Moreover for larger networks, our method yields a notable computational advantage since larger batches permit full exploitation of available GPU cores.

**Visualizing Accuracy of Parameter Gradients.** To examine the reason behind the robustness of our method, Fig. 4 visualizes the error in the final *parameter* gradients used to update the model. Specifically, we take two models for CIFAR-100—at the start and end of training—and then compute gradients for a 100 batches with respect to the convolution kernels of all layers exactly, and using our approximate strategy. We plot the average squared error between these gradients. We compare this approximation error to the "noise" inherent in SGD, due to the fact that each iteration considers a random batch of training examples. This is measured by average variance between the (exact) gradients computed in the different batches. We see that our approximation error is between one and two orders of magnitude below the SGD noise for all layers, both at the start and end of training. So while we do incur an error due to approximation, this is added to the much higher error that already exists due to SGD even in exact training, and hence further degradation is limited.

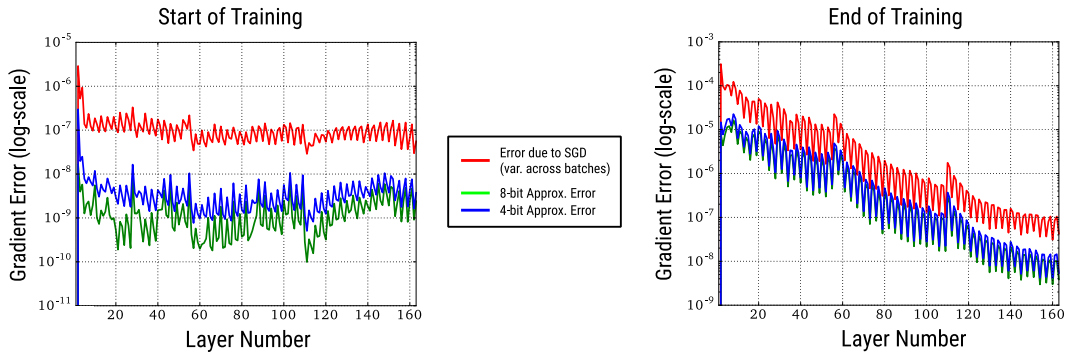

Figure 4: We visualize errors in the computed gradients of learnable parameters (convolution kernels) for different layers for two snapshots of a CIFAR-100 model at the start and end of training. We plot errors between the true gradients and those computed by our approximation, averaged over a 100 batches. We compare to the errors from SGD itself: the variance between the (exact) gradients computed from different batches. We find this SGD noise to be 1-2 orders of magnitude higher, explaining why our approximation does not significantly impact training performance.

## 5    Conclusion

We introduced a new method for implementing back-propagation that is able to effectively use approximations to significantly reduces the amount of required on-device memory required for neural network training. Our experiments show that this comes at a minimal cost in terms of quality of the learned models. With a lower memory footprint, our method allows training with larger batches in each iteration and better utilization of available GPU resources, making it more practical and economical to deploy and train very deep architectures in production environments. Our reference implementation is available at `http://projects.ayanc.org/blpa/`.

Our method shows that SGD is reasonably robust to working with approximate activations. While we used an extremely simple approximation strategy—uniform quantization—in this work, we are interested in exploring whether more sophisticated techniques—e.g., based on random projections or vector quantization—can provide better trade-offs, especially if informed by statistics of gradients and errors from prior iterations. It is also worth investigating whether our approach to partial approximation can be utilized in other settings, for example, to reduce inter-device communication for distributed training with data or model parallelism.

## Acknowledgments

A. Chakrabarti acknowledges support from NSF grant IIS-1820693. B. Moseley was supported in part by a Google Research Award and NSF grants CCF-1830711, CCF-1824303, and CCF-1733873.

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
