[Reviews · NeurIPS 2019]

Reviewer 1



- Originality: The proposed method is simple and maybe novel. I haven't seen papers that propose to do a simple uniform quantization in the preactivations just to save memory during training. But I could be over looking some of the network quantization literature. - Quality: The experiment is very thorough and solid. It shows that the proposed method is able to save memory while maintaining the same accuracy on a selection of networks on CIFAR and ImageNet. - Clarity: The paper is clearly written. I was able to understand the core contribution and Figure 2 is very nicely designed. I think it would make it clearer, if the text can explain Eq. 9 better. In my understanding, I think this step is to uniformly quantize the activations within 3 standard deviations, but this requires some extra thinking and is not immediately clear. - Significance: My major concern with the paper is regarding to the significance of this work. It is acknowledged that the proposed method can be readily implemented in lots of network architecture, thus it has good significance in terms of applications. However, the literature covered in this paper mainly focuses on memory saving, with a little on quantization. In [1] (full reference at the bottom), which is a paper published at NeurIPS last year, the authors show that it is possible to do 8-bit training and quantization together. The difference of outcome, in my opinion, is that while that work does quantization on the forward pass, the weights, and the backward pass of activations, this paper does quantization only on the backward pass of activations. Although memory saving was not a major selling point in [1], it does seem like a by-product. If I am correct, what makes this paper a separate contribution rather than a simplified version of [1]? I could be wrong so I would like to see more how this proposed method is compared to [1]. An experiment on the proposed method vs. the range BN and angle quantization scheme proposed in [1] in a more equal & controlled setting. Overall, I think this is a solid paper, but we need to see more comparison and discussion on the network quantization literature to evaluate how significant the contributions are. Therefore my overall score is 5 (marginally below). Reference: [1] Ron Banner, Itay Hubara, Elad Hoffer, Daniel Soudry. Scalable Methods for 8-bit Training of Neural Networks. NeurIPS 2018. --- Update reading the rebuttal and discussion with other reviewers, I decided to change the score to 6 to reflect the merits of the paper in terms of its simplicity and a strong experimental section.

Reviewer 2



Summary ======= This paper proposes a method to reduce the activation memory requirements needed to train (convolutional) neural networks, by using full-precision floating point numbers to compute exact activations in the forward pass, but storing quantized, low-precision versions of the activations in memory for use in the backward pass. This approach is evaluated by training ResNet models on CIFAR-10, CIFAR-100, and ImageNet, and comparing against the naive approach of using 8-bit activations for both forward and backward passes. The experiments demonstrate that it performs on par with using exact activations, while requiring ~8x less memory for activations. Originality =========== + While the method itself is very simple, the paper does a good job contrasting it to previous approaches that trade off memory savings for additional computation (e.g., reversible models) and approaches that train low-precision/quantized networks where both the forward and backward passes use 16-bit/8-bit floats. Quality ======= + The experimental results are convincing: the paper shows that using approximate activations in the backward pass closely matches both the training loss and test accuracy of exact activations, while saving ~8x activation memory. + I appreciate that the experiments are all presented with error bars. + In addition, Table 2 is very nice, as it demonstrates how this method can allow the use of significantly larger mini-batches on a single GPU. * It may be worthwhile to have a plot like Figure 3 where the x-axis is wall-clock time, to underscore the idea that the proposed method is not computationally expensive. * It is strange that the paper focuses only on convolutional neural nets, when the approach should in theory be applicable to other architectures/tasks as well. * Should include some diversity in the experiments, for example using VGG or DenseNet architectures. - Is it the case that this approach would not reduce the memory requirements of training a DenseNet, because the "outstanding layer activations that remain to be used" include all preceding layers in the network? * Most importantly, it would be good to include a discussion and evaluation of different nonlinearities like sigmoid or tanh, beyond what is already discussed in Section 3.4. Would this method work to reduce memory requirements for training RNNs as well, where the LSTM equations make use of both sigmoid and tanh activations? This would be an interesting experiment to show, or perhaps to illuminate what the failure modes of the method are. * Similarly, experiments involving Transformer networks would be useful to demonstrate broader applicability. * All the current experiments use SGD with momentum as the optimizer; it would be nice for completeness to have results using other optimizers (such as RMSprop or Adam), to understand how they perform with gradients computed using approximate activations. Clarity ======= + The paper is generally well-written. + Figure 1 is nice and conveys the method well. - However, Figure 2 is very crowded and hard to read. Significance ============ + The method has negligible computational overhead while providing a constant factor reduction in memory usage for CNNs. I think this could be a useful trick to incorporate into deep learning packages. Minor Points ============ * An interesting optional analysis would be to use the approximate activations in the forward pass, and then use exact activations in the backward pass (by storing the exact activations anyway), to verify that the exact activations are more important for computing the loss than for computing the gradient. Post-rebuttal ========== I thank the authors for their clarifications in the rebuttal. While the proposed approach is targeted towards a specific class of architectures (those with ReLU activations), I think it would be a useful trick to incorporate into deep learning frameworks. I encourage the authors to add a discussion of its limitations to the paper (i.e., that due to the sigmoid/tanh activations, it is not suited for RNNs), but I think the paper presents an interesting approach for reducing activation memory requirements of modern CNNs. As R1 mentioned, the authors can also reference and discuss Banner et al. 2018. I raised my score from 5 to 6.

Reviewer 3



Update after authors' feedback: Thank you for taking the time to answer our reviews and comments. I feel that there will be interesting bits in the updated version. Nonetheless I still think the experimental part, as well as the limitation to one type of architecture is to light for this paper to be considered as a top 50% one. I'll keep my score of 7 as it looks like an easy and efficient method to save memory in many CNN architectures. ------------------------------------------------------------------------------------------- This paper is extremely easy to read and to follow. The presented method is at the same time very simple and clever, and seems to work pretty well. It is nice to see that kind of paper, that are not overly complicated but present interesting contribution to an actual and relevant problem. The figures are a nice addition to the paper, they're clear and self-explaining (nb. I did not see Fig. 1 referenced anywhere in the text though). However, it is a shame that this method is only limited to architectures made only of convolutions, batch normalization and ReLU layers. Even though it covers a large family of models, and the authors point it out in the paper, it would have been interesting to see different architectures or applications. The experimental section is clear and interesting, the experiments seem well-designed and the results are good. An empirical comparison to other methods for saving memory, or to state-of-the-art results would have been quite valuable. That section is a bit disappointing and merely present the parameters of the models and describes the result figures and tables. Nonetheless I believe this is a solid paper and a good NeuriPS contribution. Minor points: - Beginning of Sec. 3.1: "A neural network is composition of linear" -> is a composition - end of p6 "back-ward pass", "backward-pass" -> backward pass - end of 3.4: "K = 8,4 bits" : might be clearer as "K = 4 or 8 bits"

[Author Response · NeurIPS 2019]

We thank all reviewers for their thoughtful comments, and respond to specific concerns below!

**REVIEWER 1:** Note that the submission already discusses papers that carry out quantization both in the forward and
backward pass ([6,15], in Lines 87-96 of our related work section). We will add the Banner et al. work to this discussion,
which presents a newer 8-bit quantization scheme (at the cost of some drop in performance). We note that while our
work is related, our contribution is not the quantization: indeed, we just use a simple fixed-point quantization scheme.

Instead, our contribution is in our backprop **algorithm**. As we discuss below, we believe this algorithm is both **novel**
and **significant** because it limits the accumulation of error from quantization, while still delivering savings in memory.

**The algorithm is novel** because backprop has thus far always been implemented the same way—by using the same
version of activations for forward and backward passes. Attempts at quantization have adopted this implementation and
operated "locally", by only modifying per-layer operations. Breaking from this, our algorithm takes a look at the entire
computation process in backprop, and shows it is beneficial to perform the forward pass exactly during training, and
that it is possible to do so while still saving memory. As our analysis shows, this minimizes the effect of approximation
error by preventing a cascading effect of errors building up from layer to layer.

**The algorithm is significant** because it allows the use of much higher rates of approximation and quantization, and
thus greater memory savings. Note that in our experiments, we show that even 4-bit fixed point quantization allows
successful training, even though we're using perhaps the simplest quantization function. This is precisely because we
are able to do the forward-pass in full-precision. As our experiments show, doing the same quantization naively in both
forward and backward passes simply fails.

In summary, the new algorithm prevents per-activation approximation errors from propagating across layers in deep
networks. It allows greater levels of memory savings with even simple quantization schemes, and we believe will allow
greater flexibility in exploring new kinds of approximation and quantization schemes for training.

**REVIEWER 2: - Effect of Activation Functions:** Our algorithm is designed specifically for RELU-like activations
whose gradient depends on the sign of the activation, but not the value. This is because otherwise, we would incur
additional errors while computing the gradient to input layers (eq 6), which would cause errors to build up during the
backward pass. Currently, it can't be applied to layers with sigmoid / tanh activations.

**- Applied to RNNs:** Our method can be applied to networks that have occasional sigmoid-like activations by just
leaving those layers un-approximated (e.g., we don't approximate the last softmax layer in our current experiments).
But RNNs and transformer networks have sigmoid/tanh activations in nearly every layer, and so the current version of
our method would not work on these.

We realize that this is a bit disappointing. But as R3 also points out, ReLU-based networks cover a very large class of
architectures that are widely used in many application domains. Our method will thus have real practical impact for
many researchers and practitioners who train such kinds of models. Also, we believe our method can be a starting point
for future work that targets RNNs, etc. Thus, we think this paper will be of interest to the NeurIPS audience.

**- Densenet:** If a network is fully-dense (every layer connects to every preceding layer), then our method would offer no
savings. But note that DenseNets typically have sequences of dense blocks (with dense connections within blocks), and
so $W$ would be the size of the block, not the size of the entire network. Other networks use skip connections, but only
from a sparse subset of layers, and would thus also have $W \ll L$ and allow for significant memory savings.

**- Other optimizers:** We chose momentum because this was the optimizer used by the baselines. Our method works
equally well with Adam and RMSProp. As additional analysis in the revision will show (see response to R3 below), this
is because the errors in gradients due to our approximation are much lower than from the randomness of SGD itself.

**REVIEWER 3:- More Analysis:** We'll add visualizations that give a deeper explanation of why our method works: in
addition to just showing training accuracy, we have computed results for the errors (when using our method vs exact
training) in the actual gradients of individual layer weights. We will plot these, and compare them to the error due to
SGD itself—i.e., the variance in the same gradients when computed on different mini-batches for the same model. This
will show that our approximation errors are an order of magnitude lower than SGD variance, and help demonstrate why
our approximation enables accurate training.

**- Other memory-saving methods:** Note that the main prior method for memory saving during training is checkpointing.
This is equivalent to exact training in terms of accuracy: the disadvantage being that it's slower (the memory-speed
trade-off can vary based on how frequently layers are recomputed). The other most common approach is to simply
quantize / approximate activations as and when they're computed. We compare to this strategy as our 'naive quantization'
baseline (for equivalent quantizations, we show this simply doesn't work).

[Meta-Review · NeurIPS 2019]

This paper proposes the elegant and obvious-in-retrospect idea of using exact activations for the forward pass and low-precision activations for the backward pass, thereby achieving nearly the full memory savings of low-precision activations. It shows that this scheme nearly matches the exact training curves while allowing 4-bit precision. Overall, the paper is well-executed. The writing is clear, references to related work are pretty complete, and the experiments seem sensible and convincing. The reviewers feel like the paper could have been more ambitious in certain respects (e.g. experiments on more diverse architectures), but didn't spot any major problems. The method seems genuinely useful and is probably simpler than other quantization methods. I think it should be accepted. After reading the author feedback, I'm still a little confused why the method is inapplicable to logistic activations; it's not obvious to me a priori that quantized activations couldn't give reasonably good approximations to backprop. It seems worth at least running the experiment, even if the results turn out negative. (I see no reason to delay publication over this point, though.)